# Social Determinants of the Non-Utilization of the Supplementary Feeding Program (PACAM) Aimed at Older Adults’ Nutritional Support

**DOI:** 10.3390/ijerph192114580

**Published:** 2022-11-07

**Authors:** Sandra Alvear-Vega, Héctor Vargas-Garrido

**Affiliations:** 1Faculty of Economics, University of Talca, Talca 3462227, Chile; 2Faculty of Psychology, University of Talca, Talca 3462227, Chile

**Keywords:** older people, social determinants of health, nutrition, malnutrition, frailty, risk factors

## Abstract

Chile has implemented the PACAM program to support older people with nutrition and for the prevention of malnutrition and frailty. This work aims to identify the social determinants of older persons not withdrawing PACAM food in order to obtain helpful knowledge for improving the program. First, the CASEN Survey 2017 was used (960,498 observations); the inclusion criterion was PACAM recipients (Yes/No). Next, a probit model was performed with a dichotomous response to determine the marginal effects of each independent variable (e.g., demographic, health, and social). The model shows a good fit (64.4%) with an explained variance between 10.5% to 14.1%. Those variables with more significant marginal effects are people aged 70–75, having tertiary and secondary education, urban living, not participating in social organizations, immigrants, and living in the austral zone. On the other hand, a higher likelihood of consumption was found among people of greater vulnerability (lowest income, lowest education, low health insurance, and aged over 80) and, therefore, in greater fragility. To conclude, the program achieves effective targeting, although improvement actions are required to expand coverage in some groups (indigenous people, immigrants, and people with disabilities). Moreover, authorities should evaluate and reinforce the program with tailored strategies for the older adults who actually withdraw food.

## 1. Introduction

The world’s economic and social development has brought rapid population aging [1]. The increase in life expectancy represents clear opportunities and challenges for society and older people [2]. In this context, the World Health Organization has defined healthy aging as “the process of developing and maintaining the functional ability that enables well-being in older age” (p. 28). Healthy aging assumes that intrinsic capacity and functional ability tend to decline over time; however, personal life choices and health interventions (during the lifespan) will determine each individual’s trajectory. In this vein, part of the diversity observed in older people aging is due to the genetic process associated with time; however, their physical and social environments (home, neighborhood, community, among others) are also critical, which may act as barriers or incentives in their health-related opportunities, decisions, and behaviors. Therefore, comprehensive public health programs might contribute to healthy aging [3].

Adverse health and functional outcomes attributed to poor nutrition are frequently associated with frailty [4,5]. However, among older adults, malnutrition and frailty are complex conditions that share some social determinants but have multifactorial origins [4]. Firstly, malnutrition is a common condition in older adults [3,4,6]; it can be understood as a deficient state, either due to excess or deficit of energy intake, with adverse effects on the body, functionality, or pathologies. Undernutrition is a more significant concern in older adults than overnutrition because it is associated with greater morbidity and mortality [7]. The complex aging process, along with socioeconomic factors and chronic diseases, is accompanied by weight loss, a decrease in physical activity, physiological changes (e.g., decrease in sensory capacity and secretion of hydrochloric acid, hyposalivation, chewing difficulties, among others), and a reduction in lean body mass and accumulation of body fat; all of which have an impact on nutritional status and are configured as risk factors for malnutrition [6,8]. Secondly, frailty has become a growing concern in geriatrics [4]; it overlaps and interrelates with sarcopenia [9]. Likewise, due to the problem of its definition, Fried et al. [10] have proposed that frailty corresponds to a clinical syndrome in which three or more of the following criteria are present: unintentional weight loss during the last year, exhaustion, weakness (grip strength), slow walking, and decreased physical activities. In addition, frailty implies a state of vulnerability that affects recovery after a stressful event, resulting from the cumulative decline of many physiological systems due to accumulative functioning over a lifetime [11]. Furthermore, frailty is associated with an increased risk of adverse health outcomes, including mortality, institutionalization, falls, and hospitalizations [10].

In summary, malnutrition and frailty in older people result from multiple age-related physical, psychological, and social factors [10,11]; some authors argue that there is a strong association between them, where malnutrition could be linked with sociodemographic and health conditions that lead to frailty [4]. In this line, it is proposed that adequate nutrition may contribute to healthy and frailty-protective aging [6,12,13,14]. All of the above suggests that frail older adults with malnutrition could be the target of interventions to prevent or delay adverse functional outcomes, recover their health, and improve their quality of life [6,15]. International organizations have adopted a similar approach and suggest that countries make efforts aimed at generating policies and programs that promote healthy aging [3].

The Chilean Ministry of Health has developed specific health intervention strategies, including the Supplementary Feeding Program for Older People (PACAM). The PACAM program mainly aims at public health insurance (Fondo Nacional de Salud, called FONASA) holders. Among the main objectives of this program are (a) to provide a food supplement specially designed according to the older persons’ needs; (b) to contribute to preventing and treating nutritional deficiencies in older people; (c) to contribute to maintaining or improving the physical and mental functionality of the older persons; and (d) to contribute to the detection and control of risk factors for the loss of physical and mental functionality of the older persons [16]. It delivers instant food made of cereals, legumes, or milk, fortified with vitamins and minerals, and low in total fat and sodium. The cream comes in four varieties: peas, lentils, asparagus, and vegetables (called “Golden Years” creams), and the dairy drink is based on milk and cereals, and reduced lactose (called “Golden Years” dairy drink). Masi and Atalah found fairly good acceptance of the dairy drink (organoleptic, intolerance, and subjective evaluation), similar to what was found with the creams. Likewise, these researchers underlined that unlike traditional food (rice, legumes, pasta, and others), formulated food supplements imply a higher micronutrient intake but a greater fatigue risk due to prolonged consumption [17]. By 2018, the PACAM program had a target population of 874,821 older people, and 461,321 dairy drinks and creams units were withdrawn (52.7% of coverage) [18]. Although the results of these food supplement programs raise some essential questions [19], it has also been claimed that they contribute to decreasing food insecurity and improving the health and well-being of the recipients [20,21]. In Chile, ambivalent results are also found. PACAM has been successful because it targets the most vulnerable people, its products are highly valued by society, and it has contributed to complementing the food and nutrition of older Chilean people [22], but the practical impact on the recipients has been questioned [23]. In addition, it is estimated that about 48% of the target population does not withdraw food [24]—with an impressive economic cost. These weaknesses have led authorities to foster evaluations to improve and strengthen the program [22].

PACAM provides nutritional support to older persons to prevent malnutrition and frailty, and a high percentage of the target population does not withdraw products. Therefore, the present work is aimed at identifying the social determinants associated with those older people who do not withdraw products from the distribution centers. This research aligns with the authorities’ calls to improve the PACAM [22]. In this vein, knowledge of social determinants can contribute to formulating and developing social programs which consider the conditions that could generate disadvantages in people’s health [25]. For example, some social determinants that show greater consistency are unfavorable socioeconomic conditions, age, gender, ethnicity, and demographic conditions [26]. This study will consider the data provided by the CASEN 2017 (National Socio-economic Characterization Survey) [27].

## 2. Material and Methods

### 2.1. Source of Information

This study’s data are from the National Socioeconomic Characterization Survey, CASEN 2017 [27], carried out by the Catholic University of Chile (PUC) on behalf of the Ministry of Social Development and Family. The CASEN 2017 survey’s sample units were dwellings selected in a probabilistic, stratified, and multistage manner. Inside dwellings, all family units were identified (households). CASEN 2017 was applied as a paper-and-pencil survey in a face-to-face procedure, interviewing the unit head of the family or an adult member in their absence. Data consists of a sample of 12,657 households with an older person who fulfilled PACAM recipients’ requirements. The expansion factor (EXPR) reaches 960,498 households with older adults. The EXPR allows us to obtain the results for persons and family units expanded at the national and regional levels and by area (urban and rural), representing the country’s overall population. All interviewees participated voluntarily, and anonymity is ensured under Law 17.374. CASEN 2017 is used by researchers as a secondary data source [26].

### 2.2. Outcome Variable: Withdrawal or Non-Withdrawal of PACAM Food

CASEN 2017 asked interviewees whether the older person has received or withdrawn food from the community health center or hospital—free of charge. The answer options are (1) Yes, dairy drink (milk) and Golden Years cream, (2) Did not withdraw food, or (9) Do not know/Do not recall. In this study, responses to this question were used to classify the level of use of the PACAM program: did not withdraw PACAM (1), did withdraw PACAM (0). Therefore, responses “Do not know/Do not recall” were not considered. Of the total sample in this study, using the expansion factor, 510,423 (53%) older adults withdrew PACAM products, and 450,075 (47%) did not withdraw these products.

### 2.3. Independent Variables

Independent variables were extracted according to previous findings on social determinants [24,26] as long as they were present in CASEN 2017. CASEN has seven modules (registration-residents, education, work, income, health, identities-networks participation, and housing-environment), from which three dimensions grouped the variables in this research. First, the demographic dimension (including gender, age, nationality, ethnicity, urban or rural area, and geographic location). Second is the social dimension (comprising whether the older person was household head, income, whether the older person was living alone or with company, educational level, and social organization participation). Finally, the third dimension grouped the older person’s health features (health problems, health checkups, living near/far from a community health center, and physical or mental disability). All independent variables are dichotomous, either 1 (belongs to the category) or 0 (does not). 

### 2.4. Plan of Analysis

To establish the relationship between the outcome variable “did not withdraw PACAM” and the control variables, a probit model was performed [28] with dichotomous responses. For each interaction, the average marginal effects were calculated (see Table 1). Consequently, the probit model estimated each independent variable’s marginal effect (likelihood) on the outcome variable.

### 2.5. Statistical Method

A nonlinear probit econometric probability model was used for the estimation. Probit adjusts a binary outcome variable model, assuming that a positive outcome’s likelihood results from a standard normal cumulative distribution. This model can estimate robust standard errors and then adjusts the results for complex-designed surveys, such as the CASEN 2017 used in this study [28]. Likewise, the model estimates the marginal effects (dy/dx) on the changes in the outcome variable (“no PACAM withdrawal”) triggered by a unit change in one of the independent variables taking the others as constant. The sign of the dy/dx results shows only the direction of the marginal effect: increase in probability (+) or decrease in probability (−). The standardized normal cumulative distribution function is:$$\theta (z)=\int_{-\infty }^{z}\theta (x)dx=\int_{-\infty }^{z}\frac{1}{2\pi }\exp(-\frac{1}{2}x^{2})dx$$

### 2.6. Goodness of Fit

The model was estimated based on 960,498 observations. The goodness of fit, Count R2 = 0.644, explained variance between 10.5% (Cox-Snell) and 14.1% (Nagelkerke).

## 3. Results

### 3.1. Sample Description 

With an average age of 72 years (SD = 0.55), 47% of the sample did not withdraw food from the PACAM. Of the older people who did not withdraw PACAM products, 59% were female, 51% lived alone, 99% had Chilean nationality, and 6% belonged to ethnic groups. Likewise, 51% had a good perception of their health, the same percentage that did not have an up-to-date medical checkup, 63% had had health problems in the previous three months, and 58% had a disability. On the other hand, regarding the people who withdrew the PACAM products, 61% were female, 52% lived alone, 99% had Chilean nationality, and 7% belonged to ethnic groups. In addition, 54% had a good perception of health, 69% had no medical checkup, 61% had had health problems in the previous three months, and 49% had a disability.

### 3.2. Marginal Effects

Table 1 shows the marginal effects for each control variable and the dependent variable; all the explanatory variables included in the analysis are significant (ps < 0.05).

When taking the demographic dimension, the results show that men are more likely (1.6%) than women not to withdraw PACAM products. In the 70–75 and 76–80 age groups, there is a greater probability (9.7% and 3.6%, respectively) of not withdrawing food than for people over 80 years of age. Older adults with a nationality other than Chilean are 7.9% more likely not to withdraw food from PACAM; similarly, those of native ethnic groups are more likely (1.9%) not to withdraw food from PACAM.

Concerning geographic location, an older person living more than 20 blocks (2.5 km.) from a health center has a greater probability (2.6%) of not withdrawing food from the PACAM. Likewise, if the older person lives in a rural area, the likelihood of not drawing food is lower (10.6%) than that of a person living in an urban area. Moreover, regarding the country’s regions, the people residing in the northern part of the country (regions of Tarapacá, Antofagasta, Atacama, Coquimbo, and Valparaíso) have a higher likelihood of not accessing the PACAM than a person living in the Metropolitan region (which includes the capital Santiago and is taken as the base). In the northern zone, the exception is the older adults living in the Arica region, who are less likely not to withdraw food. In addition, the older persons in the central zone (O’Higgins, Maule, Ñuble, Biobío, and Araucanía regions) are less likely to not withdraw food from the PACAM than those living in the base region. In contrast, older adults living in the southern and austral zones (Los Ríos, Los Lagos, Aysén, and Magallanes regions) have a higher possibility of not withdrawing food than the base region.

Secondly, regarding the social dimension, it was found that as the educational level increases (primary, secondary, and tertiary education), the likelihood of not withdrawing food rises (3.1%; 10.7%, and 15.5%, respectively)—considering people who have no schooling as a base. Likewise, an older person who is not the head of household has a higher probability (2.4%) of not withdrawing food than one who is. At the same time, if an older person lives alone, they have a lower probability (1.0%) of not accessing PACAM, as well as those older people who do not participate in social organizations (7.0% lower).

Regarding income, as the income quintile decreases, the probability of not withdrawing food from PACAM falls. That is, an older person belonging to the lowest income quintile (quintile I) has a lower likelihood (14.7%) of not withdrawing food from PACAM compared to older people belonging to the highest income quintile (quintile V).

Finally, in terms of the personal health dimension, older people with a bad health self-perception are less likely (1.3%) not to withdraw food from PACAM than those with a good self-perception. Consistently, older people with health problems are less likely (1.0%) not to withdraw the PACAM products, the same as older people who keep up with medical checkups in health centers (21.3%). Likewise, if the person has a disability, there is a greater probability (3.6%) of not withdrawing food from the PACAM. Regarding public health insurance, older adults classified as level a (lacking resources or migrants) and level b (earning less than the minimum wage) are less likely to not withdraw PACAM products compared to those in the highest level d (incomes above 600 dollars per month).

## 4. Discussion

Although multiple factors affect the non-withdrawal of PACAM, such as the effectiveness of communication campaigns [22] and the fatigue caused by prolonged consumption [17], this paper focuses on personal/demographic, social, and health dimensions. Among the purposes of studying social determinants is to obtain helpful knowledge that contributes to improving social programs [25]. The results of our work show some convergence with previous studies and some other new elements. All of them, taken together, could serve as inputs for reviewing and improving this program, which is the authorities’ interest [18,22]; it is aimed at nutritional security and contributing to physical and mental functionality—acting then as a buffering factor against sarcopenia [9] and frailty [16].

As a preliminary issue, there are difficulties in establishing the actual percentage of program coverage. Several factors contribute to this complexity, including seasonal withdrawal variations [16] and actual consumption by the target population [17,18,29]. The National Budget Office (NBO) has calculated coverages of 47.4% (from 2015–2018) and 59% for the year 2021 [29]. Previous statistics from the Ministry of Health indicated coverages between 77% to 65% in the years 2010 and 2014, respectively [16]. Together, these figures would suggest that the coverage percentage regarding the target population has trended downward over time. Using information from CASEN 2017, our results yielded a coverage percentage as declared by FONASA users of 53%, a figure close to that reported by the NBO for the same year [18]. In other words, there was a withdrawal trend of close to 50% of the target population, and this information was consistent with previous research using the CASEN 2015 (52%) [24]. Thus, it provides validity to our findings and, in addition, would point out that the declarations made by the recipients in the CASEN surveys were close to the coverage calculated by the NBO [18]. Furthermore, comparative data from the US shows that people aged 75 and older who participate in federal programs for nutrition support reach 62% for home-delivered and 53% for congregate meals [30]. 

We first review the personal and demographic characteristics. Official statistics indicate that PACAM withdrawal is taken by 60% of females and 40% of males [22]. If we consider that, in 2017, the total population aged 70 years and over was 41.5% males and 58.5% females [31], the figures concur with our finding that it is males who withdraw PACAM food with a slightly lower likelihood (1.6%). 

Those most in need of nutritional support are people over 80 years old with a higher frailty risk linked to dietary intake [21]. Mirroring that information, Chile observes a more significant nutrient deficit among older people as age progresses, affecting 7.9% in the 70–74 years group and rising to 16% in the group over 80 years [32]. Interestingly, our results reveal that adults over 80 withdraw food proportionally the most, while those between 70–75 years withdraw the least. This evidence would indicate that the withdrawal behavior would be proportionally adjusted to the food people’s needs. That is, the program would be adequately focused in practice, supporting the needs of the effectively interested target population. In addition, previous research has found that, despite some doubts, the consumption of PACAM products significantly increases micronutrient intake [23]. 

The detriment of people from native backgrounds regarding social program participation has been confirmed previously [26] and seems related to environmental conditions and a greater vulnerability of these communities [33,34]. Likewise, there are proven health gaps to the detriment of the migrant population [35]. Our results are in line with those antecedents.

Geographic conditions indicate that the greater the distance from the distribution centers, the lower the probability of PACAM use. First, people living more than 2.5 km away withdraw less than those living closer to the health centers. Second, the distinction between rural vs. urban population is determined by the size of the settlement, where rural population implies fewer people and, therefore, shorter distances to be traveled than urban populations (above 2000 inhabitants). Our results indicate that people in rural areas withdraw proportionally more food than those in larger urban areas. Finally, we find clear trends when considering the Metropolitan region (the country’s capital) as the base and reference point (Chile is the longest and narrowest country in the world from north to south). People in the north withdraw less from the PACAM food program (except Arica), older people in the center-south withdraw more, and people in the south and austral zones withdraw less (especially in the austral zone). If we consider the population density of those areas (north zone 26.9%, south-central zone 43.8%, and south and austral zone 10.4% [31]), together with the number of health centers (the north zone 21.0%, south-central zone 42.4%, and south and austral zone 18.4% [36]), we observe that the country’s south-central zone is the one with the highest population density and has the highest percentage of distribution centers. 

In contrast, the south and austral zone are where people are more distant from each other and have the smallest percentage of the distribution centers. For example, the Aysén and Magallanes regions have the lowest population densities (close to 1 inhabitant per km^2^) and the lowest probability of PACAM withdrawal (decreasing by 11.5% and 18.5%, respectively). In short, if an older person lives a greater distance from a distribution center, lives in an urban city, or in a region where people live far from each other, the probability of not withdrawing food from the PACAM will be greater. Geographic distance is a relevant determinant of greater or lesser recall. 

Second, regarding social characteristics, it must be noted that a social determinant that shows grand consistency and persistence over time is socioeconomic conditions [26]. In the current study, variables such as income quintile, educational level, and type of public health insurance are related to the socioeconomic condition of the older person. These three variables’ results were consistent; older people progressively withdraw more food as they move towards greater economic vulnerability. That is, the lower the educational level, the lower the income level, or the lower the type of public health insurance, the higher the tendency to withdraw PACAM food.

In addition, people who are heads of households and those who participate in social organizations withdraw more PACAM food. These individuals are more likely to be empowered, have greater mobility, and have the possibility of generating social networks, all of which facilitate access to information and the help of others in gaining access to food. On the other hand, older people who live in the company of others withdraw less food than those who live alone. Similarly, in Colombia, Restrepo et al. [8] found that people in family isolation consume more supplementary food than those who do not. In this sense, it is complex to arrive at a definitive conclusion. Previous works have had mixed findings, depending on who the older person lives with, as well as their cultural background. For example, in Australia, people living with their spouses have better dietary intake, while single men living in the USA have a higher nutritional risk than their European peers [37]. In this work, our results consistently show that older people with a greater vulnerability withdraw the most (except for some specific groups). Considering this information, we can hypothesize that people who live with others in Chile (i.e., not heads of household) find a protective factor in this condition. Living in the company of others makes them more nutritionally secure. Therefore, they tend to withdraw fewer PACAM products—a piece of information contributing to the food dilution debate [17]. The same approach would explain our results for people living alone, implying a greater nutritional risk and, therefore, a greater tendency to withdraw products. That is, living alone would be a factor of vulnerability or risk of frailty. Also, this explanatory mechanism fits with general reviews pointing out that people living alone are in a situation of greater vulnerability, especially those older people in developing countries [38].

Finally, one of the PACAM program’s requirements is having up-to-date medical checkups. Therefore, it is not surprising that having medical checkups is a condition that considerably reduces non-withdrawal (21%). Likewise, older people who perceive themselves as having poor general health and those having real health problems withdraw more PACAM foods (1.3% and 1.0%, respectively); people with more health problems visit the community medical centers more often, thus facilitating the withdrawal of the products. The figure that raises a red flag in health conditions is that people with disabilities withdraw fewer products proportionally (3.6%), likely due to the need for more significant assistance when withdrawing and the lack of adequate information about the program.

Among the limitations of this work are that, first, we used a survey with participants’ subjective responses about PACAM products withdrawal; some participants could have stated that they withdrew food due to this option’s social desirability. Second, this study does not consider the interactions between the control variables and their sampling weights; the results do not consider the increase or decrease in the likelihood due to a one-unit increase in other variables since they are taken as constants in the model. Also, a huge sample could increase the chance of type I error. Despite that, our results are consistent with previous antecedents, providing convergence validity for our findings. 

## 5. Conclusions

Using CASEN 2017, we found that the PACAM program covers nearly 50% of the target population. Moreover, people with higher consumption trends are in conditions of greater vulnerability (lower economic conditions, older ages, living alone) and, therefore, in greater fragility. Although doubts have been raised about its effectiveness, there is also evidence that the PACAM program strengthens nutrient intake [23]. This information coincides with international evidence that these programs help decrease nutritional insecurity, which is usually accompanied by limitations in daily activities [39]. Thus, our findings would indicate that the program achieves effective targeting since those who need it the most withdraw food the most, which likely underlies the high public valuation of the program [22]. However, improvement measures are needed to expand coverage in groups that withdraw less food proportionally, such as people of indigenous ethnicities, immigrants, geographically remote people, and people with disabilities. 

On the other hand, most older people in Chile have a normal weight [23]; this work has found that most vulnerable older people effectively access the PACAM program. Thus, along with measures to expand the program’s coverage among the vulnerable groups, the authorities could consider a more targeted strategy tailored to older persons who withdraw food. For example, the range of program options could be expanded, allowing greater autonomy, and fitting the reality of each recipient, such as considering additional options to the current creams and dairy drinks, having food boxes, cards for purchasing healthy food, and vitamin supplements in capsules, depending on the nutritional evaluations of professionals during medical checkups.

## Figures and Tables

**Table 1 ijerph-19-14580-t001:** Predict (the older person did not withdraw PACAM).

		dy/dx	[95%. Conf. Interval]
Gender (Male = 1; Female = 0)		0.016 ***	[0.01393; 0.0190]
Age	≥70 ≤75	0.097 ***	[0.0953; 0.1004]
	≥76 ≤80	0.036 ***	[0.0340; 0.0396]
	>80 (base)	-	-
Educational level	Tertiary education	0.155 ***	[0.1497; 0.1609]
	Secondary education	0.107 ***	[0.1030; 0.1110]
	Primary education	0.031 ***	[0.0277; 0.0350]
	Without (base)	-	-
Nationality (Others = 1; Chilean = 0)		0.079 ***	[0.0681; 0.0901]
Native ethnic groups (Yes = 1; No = 0)		0.019 ***	[0.0155; 0.0243]
Head of household (No = 1; Yes = 0)		0.024 ***	[0.0213; 0.0281]
Living alone (Yes = 1; No = 0)		−0.010 ***	[−0.0133; −0.0741]
Social organizations involvement (No = 1; Yes = 0)		0.070 ***	[0.0685; 0.0728]
Self-perceived general health (Bad = 1; Good = 0)		−0.013 ***	[−0.0152; −0.0109]
Health problems (Yes = 1; No = 0)		−0.010 ***	[−0.0124; −0.0080]
Medical check-ups (Yes = 1; No = 0)		−0.213 ***	[−0.2159; −0.2118]
Physical or mental disability (Yes = 1; No = 0)		0.036 ***	[0.0341; 0.0385]
Public health insurance	level a	−0.079 ***	[−0.0851; −0.0744]
	level b	−0.086 ***	[−0.0918; −0.0815]
	level c	0.009 *	[0.0018; 0.0164]
	level d (base)	-	-
Distance to a community health center (Far = 1; Close = 0)		0.026 ***	[0.0229; 0.0295]
Residence (Rural = 1; Urban = 0)		−0.106 ***	[−0.1097; −0.1028]
Quintile	I	−0.147 ***	[−0.1520; −0.1430]
	II	−0.137 ***	[−0.1417; −0.1328]
	III	−0.081 ***	[−0.0857; −0.0766]
	IV	−0.059 ***	[−0.0646; −0.0552]
	V (base)	-	-
Region	Arica	−0.088 ***	[−0.0996; −0.0778]
	Tarapacá	0.047 ***	[0.0379; 0.0575]
	Antofagasta	0.018 ***	[0.0105; 0.0272]
	Atacama	0.034 ***	[0.0253; 0.0437]
	Coquimbo	0.042 ***	[0.0365; 0.0474]
	Valparaíso	0.017 ***	[0.1435; 0.0212]
	Metropolitana (base)	-	-
	O’Higgins	−0.004 *	[−0.0093; −0.0003]
	Maule	−0.030 ***	[−0.0348; −0.0259]
	Ñuble	−0.097 ***	[−0.1032; −0.0919]
	Biobío	−0.072 ***	[−0.0756; −0.0684]
	Araucanía	−0.025 ***	[−0.0300; −0.0210]
	Los Ríos	0.052 ***	[0.0456; 0.0592]
	Los Lagos	0.014 ***	[0.0094; 0.0189]
	Aysén	0.115 ***	[0.1008; 0.1309]
	Magallanes y Antártica	0.185 ***	[0.1732; 0.1972]

* *p* < 0.05; *** *p* < 0.001.

## Data Availability

The CASEN 2017 database analyzed during the current study is available on the MSDF webpage, http://observatorio.ministeriodesarrollosocial.gob.cl/encuesta-casen-2017 (accessed on 9 September 2022).

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
