# Peer review of "Social Determinants of the Non-Utilization of the Supplementary Feeding Program (PACAM) Aimed at Older Adults’ Nutritional Support"

_ijerph, 2022, doi:10.3390/ijerph192114580_

Round 1

Reviewer 1 Report

The authors address an important topic in the use of public health food provisions for mitigating malnutrition and frailty in older adults. The researchers tested a large number of variables to identify targets of intervention. 

Comments:

1. Suggest including "sarcopenia" in your introduction and discussion as you describe frailty and malnutrition. Sarcopenia is a term commonly used in the literature when addressing frailty in older adults. Possible citation: Gingrich, A., Volkert, D., Kiesswetter, E. et al. Prevalence and overlap of sarcopenia, frailty, cachexia and malnutrition in older medical inpatients. BMC Geriatr 19, 120 (2019). https://doi.org/10.1186/s12877-019-1115-1

2. It would be valuable to compare your withdrawal/participation rates with those in other countries. Suggested reading: Rudolph C, Francis S. MEALS ON WHEELS: PERSPECTIVES FROM TODAY’S AGING ADULT. Innov Aging. 2018 Nov 16;2(Suppl 1):983. doi: 10.1093/geroni/igy031.3637. PMCID: PMC6239756.

3. Discuss the acceptance of the foods offered in PACAM and whether this could influence participation/withdrawal rates. Were the foods tested for palatability in the population of older adults?

Reviewer 2 Report

A brief summary

Considering the increase of the elderly population in the world, this topic seems extremely interesting and scientifically acceptable. Nutrition is a very important factor, both in the young and in the elderly.

General comments

The summary is well designed, but the headings (Introduction, Objectives...) should be removed.

The Introductory part is too extensive. Although all the parts mentioned by the authors are relevant, they can be significantly shortened. The Introduction should be in one paragraph, without the headings.

It is advised to replace part of the references with a newer one (within 5 years), considering that almost half of it is older.

Specific comments

Line 90                  Please explain the abbreviation FONASA (in native language), the first time it is mentioned.         

Line 125    Explain the abbreviation CASEN, the first time it is mentioned and use uppercase.

Line 142    This should be written in the past tense. For example, instead of "CASEN 2017 asks interviewers" put "CASEN 2017 asked interviewers"

Line 199-201       Please rewrite the sentence to make it clearer.

Line 218                Please add the unit to the number.

Line 223 – 234    It is suggested to move this part from the Discussion to the Introduction.

Line 241                Explain the abbreviation DIPRES, the first time it is mentioned.

Line 265-266       Please rewrite the sentence to make it clearer. The expression „the target population it is drawing “could be rewritten more clearer for example: „target population of interest “or similar, depending on what the authors want to say.

Line 276-277       Considering that some sentences from the Results are repeated in the Discussion (only in a slightly different formation), it is suggested not to repeat them, but to decide where this fact will be noted. The discussion continues by commenting on the results and comparing them with other research.

Line 300-301       Please connect those two sentences and make them clearer.

Line 305-307       Please rewrite the sentence to make it clearer.

Line 357                Suggestion: Perhaps a better term would be „improvement measures “.

Reviewer 3 Report

It is interesting study where authors investigated the Social Determinants of the Non-utilization of the Supplementary Feeding Program among old adults. Although there are some interesting findings, the technical issues should be addressed further if a revision is invited.

1.      In section of methods, authors should place a specific part on statistics to state statistical methods used in this study.

2.      About goodness of fitting, LR chi-square is huge, and P is less than 0.05, how to explain this goodness of fitting? Maybe it is due to huge sample size. In fact, the aim of this stud is to identify risk factors for not withdraw PACAM, so table 1 maybe not necessary.

3.      Authors should state clearly Probit model in the method part, including what is dy/dx in table 1? It is probability?  Seems not because there is negative values. Therefore, author should explain them carefully in the text.

4.      I suggest that authors should present the basic information on this sample at beginning of results.  

5.      Authors should interpret results with caution because there is a huge sample which increase chance of type I error.
